# Supervised Band Selection with a Concrete Layer for Hyperspectral Imagery in Remote Sensing and Autonomous Driving

## Abstract

Hyperspectral imagery captures rich spectral information, which is valuable for a wide range of applications but poses challenges due to high data dimensionality. Current band selection methods are often computationally intensive, non-embedded, or lack adaptability for specific tasks. We address this gap by introducing a novel plug-and-play embedded method for supervised band selection in hyperspectral imagery, utilizing a concrete selector layer based on the Gumbel-Softmax re-parameterization trick. Our approach allows for dynamic and task-specific selection of optimal bands, eliminating the need for pre-processing and enabling seamless integration with downstream models. We evaluated the method on four hyperspectral datasets, covering three remote sensing benchmarks and an autonomous driving task, demonstrating consistent improvements over state-of-the-art methods. This is the first work to perform comprehensive band-selection research on an autonomous driving dataset of this type, and the first to employ a concrete layer for supervised band selection. Our findings highlight the potential of this approach for real-time hyperspectral analysis in applications such as autonomous driving and environmental monitoring, laying the groundwork for further exploration of efficient, domain-specific band selection.

## 1 Introduction

Hyperspectral imaging (HSI) captures the complete optical spectrum at each point within an image. Unlike a standard color camera, which records light intensity in three colors (red, green, and blue), a hyperspectral camera captures a wide range of wavelengths, typically consisting of several hundred bands. This transition from color to full hyperspectral imaging significantly increases the amount of information captured, offering considerable potential across various applications such as medical imaging, agriculture, aerial photography, and autonomous driving (Gutiérrez-Zaballa et al., 2023; Gutiérrez-Zaballa et al., 2022; Bayramoglu et al., 2017; Gao et al., 2020; Kazi Saima Banu & Gardea-Torresdey, 2024; Sun et al., 2024).

One notable advantage of HSI is its ability to provide detailed spectral information, enabling improved differentiation between materials that might otherwise look similar in traditional RGB imaging. This capability enhances object segmentation, especially for applications like Advanced Driver-Assistance Systems (ADAS) and Autonomous Driving Systems (ADS), leading to greater accuracy and robustness in object identification and tracking tasks (Huang et al., 2021; Colomb et al., 2019; Weikl et al., 2022).

However, leveraging the full range of spectral bands provided by HSI introduces several challenges. The vast amount of data generated by HSI systems requires significant resources for sensor hardware, storage, transmission, and analysis, making these systems expensive and cumbersome. The high dimensionality of HSI data also complicates real-time processing and increases the computational burden for many applications.

Therefore, there is a critical need for effective band selection algorithms that can reduce the number of bands while retaining essential information for downstream tasks. Targeted selection of relevant spectral bands can enhance the efficiency of deep learning models, making HSI more practical for real-world applications across various domains. Improved band selection algorithms can also help in

the design of sensors that retain the benefits of HSI while integrating simpler technologies which is crucial for the advancement of intelligent ADAS/ADS (Pinchon et al., 2019; Winkens et al., 2019).

While existing research has explored band selection methods, these often involve independent unsupervised or supervised preprocessing steps, which can result in suboptimal band choices for specific downstream tasks in hyperspectral imaging (HSI).

In this paper, we present a novel plug-and-play embedded method for supervised band selection in hyperspectral imagery, which stands apart from existing techniques due to its seamless integration and efficiency. Unlike traditional methods that often require separate pre-processing or unsupervised feature selection, our approach directly integrates the band selection process into the training pipeline without additional pre-processing steps. Central to our method is the innovative use of the Gumbel-Softmax re-parameterization trick (Jang et al., 2017), which allows for differentiable, supervised selection of optimal spectral bands, enabling the model to dynamically identify the most informative features for each task. This approach uniquely combines the strengths of concrete selector layers with the flexibility to learn which bands are most relevant, enhancing both model performance and simplicity.

Our method effectively learns the optimal bands as an integral part of a Convolutional Neural Network (CNN) (O'shea & Nash, 2015), focusing on the challenging task of semantic segmentation, which predicts semantic categories for each pixel in an image. We show that our model consistently outperforms state-of-the-art band selection techniques across four hyperspectral datasets, including remote sensing and autonomous driving tasks. Notably, our method excels even when selecting a small number of bands, highlighting its practicality for designing low-cost, deployable sensors suitable for real-world applications.

## 2 RELATED WORK

Recent advances in hyperspectral band selection can be categorized into several key approaches, distinguished by their supervision level (supervised vs. unsupervised) and integration with downstream tasks (embedded vs. non-embedded).

Unsupervised and non-embedded methods are often applied as a pre-processing step without tailoring to or integration into downstream models. Dimensionality reduction based methods include PCA-based systems such as Kang et al. (2017). Reconstruction-based techniques, such as BS-nets (Cai et al., 2020), DARecNet-BS (Roy et al., 2020), and TAttMSRecNet (Nandi et al., 2023), frame band selection as a reconstruction problem using autoencoders. Sparsity-based methods, like SpaBS (Sun et al., 2014) and SNMF (Sun et al., 2015), aim to find sparse representations and often utilize clustering.

Supervised non-embedded techniques use labeled data to guide the selection of the most informative spectral bands but are not directly integrated into the downstream model during training. Instead, they perform the band selection as a separate pre-processing step. Genetic algorithms such as Ou et al. (2023); Esmaeili et al. (2023) select predictive bands via an evolutionary process. Concrete Autoencoders have also been applied for unsupervised band selection by leveraging concrete random variables and reconstruction loss to select optimal bands based on an information entropy (IE) criterion (Abid et al., 2019; Sun et al., 2021). Deep Reinforcement Learning (Mou et al., 2021; Feng et al., 2021; 2024), maximize the utility of selected bands for specific tasks.

Regularization-based methods such as EHBS (Zimmer & Glickman, 2024) stand out as some of the few supervised and embedded approaches, enabling efficient integration with the learning pipeline by imposing constraints (e.g., relaxation of the $l_0$ norm).

Despite these advancements, state-of-the-art (SOTA) methods still primarily focus on remote sensing applications and are often not optimized for selecting a small number of bands. Moreover, most existing techniques are non-embedded, complicating integration with downstream models and leading to suboptimal accuracy. To the best of our knowledge, this work is the first to present a comprehensive, supervised, and embedded band selection approach that utilizes a concrete layer with the Gumbel-Softmax re-parameterization trick, demonstrating its effectiveness on both remote sensing and autonomous driving datasets, thereby setting a new benchmark for band selection in hyperspectral imagery.

## 3 METHOD

### 3.1 PROBLEM DEFINITION

Let $X$ represent a sample of $m$ data instances where each instance is an $n$-sized array of 2D images and $Y$ denotes the corresponding $m$ labels. Here, $n$ represents the total number of available spectral bands.

Let $F$ be a family of models for the downstream task, each accompanied by a choice of parameters $\theta$, and $Loss$ is a loss function between a specific label $y_i$ and a corresponding model output $\hat{y}_i$.

We denote a possible band selection using an indicator vector $\mathbf{I} \in \{0,1\}^n$, where $\mathbf{I}_j = 1$ if band $j$ is selected for processing, and $\mathbf{I}_j = 0$ if it is not. The $l_0$ norm $\|\mathbf{I}\|_0$ of an indicator function $\mathbf{I}$ corresponds to the number of selected bands. We denote $x \odot \mathbf{I}$ as the point-wise product between an input item $x$ and the indicator vector $\mathbf{I}$ in which all non-selected bands are effectively masked to zero. Let $k < n$ be the target number of bands.

The goal of embedded band selection methods is to simultaneously select an indicator vector $\mathbf{I}$ and a model $f_\theta \in F$ that minimize the overall loss of the data as follows:

$$\arg \min_{\theta, I, \|\mathbf{I}\|_0 = k} \frac{1}{m} \sum_{i=1}^{m} (Loss(f_\theta(x_i \odot \mathbf{I}), y_i)). \tag{1}$$

### 3.2 PROPOSED FRAMEWORK

Our proposed system is an end-to-end embedded system. It comprises a downstream task model enhanced with an additional concrete selector layer inserted between the input and task models. This added layer is based on the principles of the Gumbel-Softmax trick (Jang et al., 2017). Our novel adaptation is specifically tailored for supervised hyperspectral band selection within the context of image semantic segmentation. By adding this layer, our proposed model leverages the intrinsic characteristics of hyperspectral data and addresses the unique requirements of semantic segmentation tasks, seamlessly integrating with downstream models without the need for band-selection prepossessing.

In the setting of band selection and contrast to the standard feature selection setting, all features of a given band should either be included or excluded from the input. We have thus adapted the concrete layer, initially designed for feature selection, to work over groups of features. This is done by altering the gates layer to either mask all the features in the group (i.e., band) or to leave the features intact. This layer comes right after the input layer and precedes the downstream task's first layer of the deep learning network.

### 3.3 CONCRETE SELECTION LAYER

The concrete band-selection layer uses the Gumbel-Softmax trick to create a differentiable approximation of discrete feature selection. This layer is embedded within a neural network between the input layer and the downstream task model to allow the model to learn which features to select during training. The concrete band-selector layer is defined by $(L, \tau, \alpha, \beta)$. $L$ is a learnable $k \times n$ logits matrix in which $n$ is the total number of bands and $k$ is the target number of bands. Each $n$-dimensional row of $L$ corresponds to a selector for a specific band. $\tau \in (0, \infty)$ controls the temperate parameter of the corresponding concrete distributions. The temperature $\tau$ is initialized with a certain value and is gradually reduced by the decay factor $\alpha$. As the temperature gradually lowers towards 0, smoothly shifting from exploration to exploitation, the concrete random variables approach the discrete distribution, outputting one-hot vectors and thus acting as a band-specific mask on the input. $\beta$ is the scale parameter of the Gumbel distribution that controls the magnitude of the noise added to the logits when calculating the mask vectors in the forward pass.

In each forward pass, the calculated Gumbel softmax per each row of the matrix is multiplied by the full HSI 3D input and is passed on to the first layer of the deep learning network of the downstream task. Given an input tensor $\mathbf{X} \in \mathbb{R}^{n \times s_1 \times s_2}$, where $n$ is the number of bands and each band is an image of size $(s_1, s_2)$, the encoder encodes $\mathbf{X}$ into a $\mathbb{R}^{k \times s_1 \times s_2}$ tensor by multiplying it by a matrix $\mathbf{M} \in \mathbb{R}^{k \times n}$. The matrix $M$ is derived from the logit matrix $L$ and the temperature parameter $\tau$ by

applying the Gumbel-Softmax operation over the rows of $L$ as follows:

$$M_{i,j} = \frac{\exp((L_{i,j} + G_{i,j})/\tau)}{\sum_{r=1}^{n} \exp((L_{i,r} + G_{i,r})/\tau)},$$

where $G_{i,j} = -\log(-\log(u_{i,j}))$ and $u_{i,j} \sim \text{Uniform}(0, \beta)$ and $\beta$ is the scale parameter of the Gumbel distribution that controls the magnitude of the added noise.

The values of $(L, \tau, \alpha, \beta)$ need to be initialized before training begins, as they play a critical role in the learning process of the model. The values of the matrix $L$ are learned as part of the training process of the entire neural network in an attempt to solve the optimization problem defined in equation 1. As $\tau \to 0$ ( by multiplying it by the decay factor $\alpha$ at the end of each batch), the distribution becomes more discrete, whereas larger values of $\tau$ result in a smoother, more probabilistic selection. This re-parameterization allows for differentiable sampling, enabling gradient-based optimization during the training process. During inference, only the Gumbel-Softmax is not applied but rather a corresponding one-hot encoded matrix derived from the logits matrix is used to get a final selection in an efficient and deterministic manner.

## 4 EXPERIMENTAL SETTING

This section outlines the experimental settings used to evaluate our proposed model across the four datasets and two distinct tasks discussed earlier. Since the characteristics and requirements of the remote sensing and autonomous driving datasets vary, we adapted the experimental procedures accordingly.

### 4.1 DATASETS AND TASKS

We evaluated our proposed model on four semantic segmentation datasets: three remote sensing datasets—Pavia, Salinas, and Chikusei (Pavia; Salinas; Yokoya & Iwasaki, 2016)—and HSI-Drive V2, an Autonomous Driving Systems (ADS) dataset (Gutiérrez-Zaballa et al., 2023). Table 1 summarizes the properties of these datasets.

Table 1: Datasets Summary

| Type | Dataset | Bands | Spectrum (nm) | Classes | Samples | Notes |
|---|---|---|---|---|---|---|
| Remote Sensing | Pavia | 103 | 430-860 | 9 | 42,776 | |
| | Salinas | 204 | 430-2500 | 16 | 54,129 | Each sample is a pixel |
| | Chickusei | 128 | 363-1018 | 19 | 5,877,195 | |
| ADS | HSI-Drive V2 | 25 | 598-976 | 10 | 756 | Each sample is an image |

The Pavia dataset captures an urban scene in northern Italy, providing 103 spectral bands in the 430-860 nm range, with approximately 42,000 valid pixels classified into nine categories. The Salinas dataset, focused on agricultural scenes in California, contains 204 bands (after removing water absorption bands) in the 430-2500 nm range, with approximately 54,000 valid pixels and provides 16 ground-truth classes. It is known for its high spatial resolution (3.7-meter pixels). The Chikusei dataset is a large-scale benchmark covering urban and agricultural areas in Japan, with 128 spectral bands in the range of 363 to 1018 nm and annotations for 19 classes. Figure 1a displays a sample area of the scene in grayscale alongside the corresponding color-coded ground-truth annotation.

The HSI-Drive V2 dataset contains 756 images across 25 spectral bands, labeled into 10 classes and collected under diverse weather conditions. This dataset is intended to evaluate scenarios relevant to autonomous driving, such as low lighting and rainy weather, providing a robust benchmark for real-world ADS tasks. We used the simplified 5-class grouping task focusing on 5 key classes (road, road marks, sky, vegetation and other) for enhanced separability as this was the setting for which classification results were reported in the dataset paper. Figure 1b shows an example image from the dataset along with the corresponding annotated segmentation.

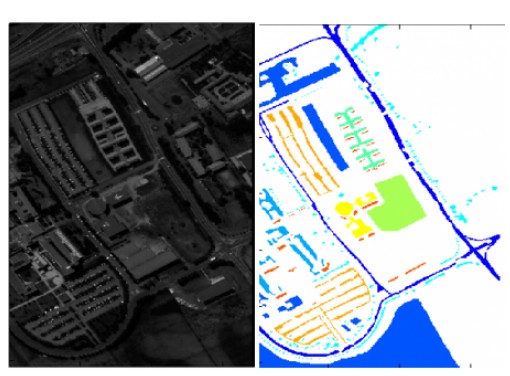 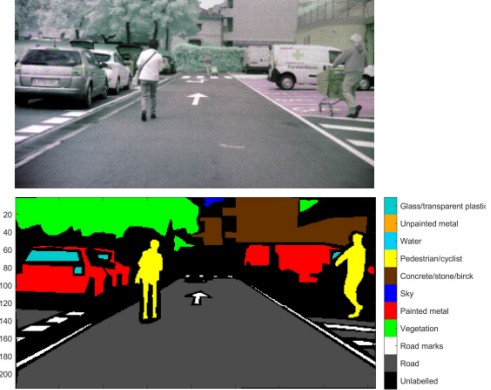

(a) Pavia University scene: (left) A sample band of the scene in grey scale, and (right) the corresponding color-coded annotations of the image pixels according to the different target classes.

(b) Hsi-Drive V2 dataset example image: (top) raw RGB image, and (bottom) coresponding color coded object segmentation annotations.

Figure 1: Example Scenes from PaviaU and Hsi-Drive V2 datasets

## 4.2 EXPERIMENTAL SETTINGS FOR THE REMOTE SENSING TASK

For the remote sensing datasets (PaviaU, Salinas, and Chikusei), we employed a 10-fold cross-validation scheme where the pixels were randomly split into training and validation folds. Initial hyperparameter tuning was conducted using the Pavia dataset, and the resulting settings were then applied to the Salinas and Chikusei datasets.

For the downstream classification task, we implemented a 3D Convolutional Neural Network (CNN) model, following the architecture outlined in Ben Hamida et al. (2018). This model classifies small patches of a hyperspectral image (HSI) cube by assigning the label of the central pixel, which is a common practice in the field (Chen et al., 2016; He et al., 2017; Luo et al., 2018; Ben Hamida et al., 2018; Lee & Kwon, 2016; Paoletti et al., 2023; Giri et al., 2024; Zhang et al., 2024). The network consists of three 3D convolutional layers, operating on input patches of 5x5 pixels, followed by a fully connected linear layer. We used a batch size of 256 and employed cross entropy as the loss function

For evaluation, we used overall accuracy as our main metric. We also measured average accuracy and Kappa score as they are commonly used to evaluate this task (Nandi et al., 2023; Cai et al., 2020).

## 4.3 EXPERIMENTAL SETTINGS FOR THE AUTONOMOUS DRIVING TASK

For the autonomous driving dataset (HSI-Drive V2), which involves image-level tasks, we split the 756 images into training, validation, and test sets in a 7:1:2 ratio. Hyperparameter tuning was performed using the validation set, and results were reported on the unseen test set. We followed the experimental setup proposed by the original authors of the dataset, conducting one experiment for a 5-class scene understanding Advanced Driver Assistance Systems (ADAS) task. For the downstream task models, we implemented a UNet-based architecture that classifies the HSI image into a semantic mask as is commonly done for this task (Long et al., 2015; Gutiérrez-Zaballa et al., 2023). The encoder consists of 4 convolutional blocks, each applying a 0.5 downsampling factor to compress the input data. Each block utilizes a kernel size of 3 for the convolutional operations. The decoder mirrors this process by progressively upsampling the data to restore the original dimensions. As the loss function, we used Weighted Cross Entropy.

For the drive dataset, as each sample is an image, we have used, in addition to Overall and Average Accuracy, Average IOU, Precision, and Recall per each label class, as reported in similar settings. (Gutiérrez-Zaballa et al., 2023).

## 4.4 BASELINE BAND SELECTION METHODS

To compare our proposed method to other state-of-the-art methods, we implemented nine different band selection methods appearing as top-performing band selection methods in recently published work covering the different family types described in section 2. The list of methods used for comparison is summarized in Table 2. All methods were used over the PaviaU and Salinas remote

| Method | Family | Reference |
|---|---|---|
| Gumbel (Ours) | Embedded encoder | |
| SNMF | Sparsity and Clustering | Sun et al. (2015) |
| Genetic | Supervised genetic optimization | Shaw (2020) |
| BS-Net-Conv | Autoencoder Reconstruction | Cai et al. (2020) |
| TAttMSRecNet | Autoencoder Reconstruction | Nandi et al. (2023) |
| DARecNet-BS | Autoencoder Reconstruction | Roy et al. (2020) |
| DRL | Reinforcement Deep Learning | Mou et al. (2021) |
| PCA | Dimension Reduction | Sun & Du (2018) |
| SpaBS | Sparsity | Sun et al. (2014) |
| EHBS | Regularization Deep Learning | Zimmer & Glickman (2024) |

Table 2: Baseline band selection methods

sensing datasets. For the two larger datasets (Chickusei and HSI-Drive V2), we limited our comparison to the top 3 methods that performed best on the PaviaU and Salinas datasets (appearing at the top of Table 2) while still covering the three prominent families: SNMF-sparsity and clustering, BSNETS-Reconstruction with Deep Learning, and Genetics-Supervised extensive searching with genetic optimization.

## 4.5 INITIALIZATION

As part of the implementation of the Concrete Selection Layer (see section 3.3), one needs to initialize the model parameters $(L, \tau, \alpha, \beta)$. The temperature params $(\tau, \alpha)$ and the noise param $(\beta)$ were treated as hyperparameters, and their initialization values were tuned in the same manner as other network hyperparameters as described above. We tested different initial temperatures of $\tau \in [0, 10]$ and $\alpha \in [0.99, 0.99999]$. The values chosen and used for the final evaluation of the remote sensing datasets were $\tau = 1.5$, $\alpha = 0.99998$, $\beta = 0.15$, and for the drive dataset $\tau = 8.5$, $\alpha = 0.9999$, $\beta = 0.15$.

As for the logits matrix $L$, we tested two different initialization schemes. The naive initialization consisted of random initialization of the complete matrix in a uniform manner. In this scheme, each row of $L$ was initialized using Xavier (Glorot) initialization, in which the values are drawn from a distribution with mean 0. In addition, we tested a novel initialization scheme in which we tried to head-start the selectors, each on a different range of the spectrum. To achieve this, given an $k \times n$ logits matrix $L$ in which $n$ is the total number of bands, and $k$ is the target number of bands, we segment the $n$ bands into $k$ segments each of size $\lfloor \frac{n}{k} \rfloor$. Each row (gate) in $L$ is initialized to focus on different segments. It is done by adjusting the Xavier initialization in such a way that the mean value of the area we want the gate to focus on to a positive is positive and negative for the other row values while maintaining an overall mean (0) and variance as in the naive Xavier initialization.

An example of such an initialized matrix when choosing 5 out of 25 bands is illustrated in Figure 2. Our experimentation showed that our novel initialization scheme produced better results and avoided the concrete selection layer from selecting duplicate bands. We thus used the improved initialization scheme in our final settings, for which we report our results. We discuss this further in section 6.

## 4.6 COMPUTE AND REPRODUCABILITY

Experiments were done on an NVIDIA GeForce GTX 1080 Ti instance, with code written in Python 3.8 using PyTorch Paszke et al. (2017) version 2.2.2. The drive dataset requires 400MB of memory, and the Remote sensing datasets require 1GB. Each epoch takes 1-5 minutes, and the number of epochs is reported in the experiment settings. The code and implementation details for this work

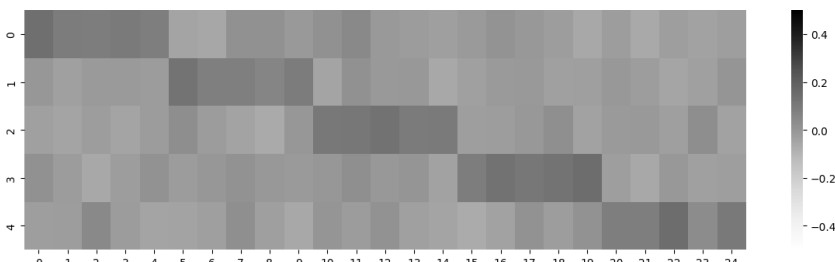

Figure 2: Heatmap of the logits matrix $L$ after applying the novel initialization scheme, illustrating the segmentation of 25 bands into 5 target bands, where each selector is biased towards a specific spectral range. This initialization helps prevent the selection of duplicate bands by providing a head-start for each gate.

will be made publicly available on GitHub, with the repository link provided in the final version of this paper.

## 5 RESULTS

### 5.1 REMOTE SENSING RESULTS

The overall accuracy results comparing our method to various baseline methods for different numbers of selected bands on the Pavia and Salinas datasets are shown in Figure 3. Our proposed Gumbel-based method consistently outperforms the other methods for nearly all target numbers of bands in both datasets and achieves the highest AUC results, as also shown in Figure 3.

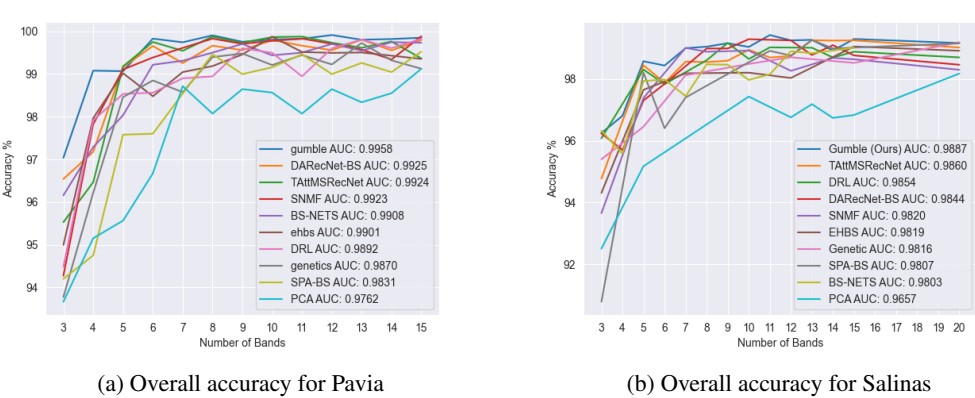

(a) Overall accuracy for Pavia       (b) Overall accuracy for Salinas

Figure 3: Overall accuracy and AUC results comparing our Gumbel-based method to other band selection methods for different numbers of selected bands over (a) the Pavia dataset and (b) the Salinas dataset.

Table 3 shows detailed overall accuracy, average accuracy, and Kappa results over the Pavia dataset for 3 and 8 bands selected. As highlighted in the table, our method outperforms all other methods in all 3 metrics when 8 bands are selected and outperforms all other methods for Kappa and overall accuracy when 3 bands are selected. For average accuracy on 3 bands our method is second best to BS-Net-Conv with very similar results.

Overall accuracy and AUC results for the Chikusei dataset are shown in Figure 4a. We have tested our method and compared it to the top performing band selection methods on the PaviaU and Salinas datasets. Results show that our method outperforms all other methods in overall accuracy on a low number of selected bands as well as in the overall AUC.

| Method/Metric | 3 bands | | | 8 bands | | |
|---|---|---|---|---|---|---|
| | Kappa | OA | AA | Kappa | OA | AA |
| Gumbel (Ours) | **0.97 ± 0.01** | **97.41 ± 0.46** | 97.35 ± 0.63 | **0.9987 ± 0.0006** | **99.90 ± 0.048** | **99.92 ± 0.58** |
| EHBS | 0.93 ± 0.02 | 94.28 ± 1.65 | 94.49 ± 1.1 | 0.9893 ± 0.006 | 99.185 ± 0.459 | 99.43 ± 0.42 |
| genetic | 0.93 ± 0.01 | 94.59 ± 0.64 | 94.86 ± 0.99 | 0.992 ± 0.003 | 99.39 ± 0.26 | 99.54 ± 0.15 |
| TAttMSRecNet, | 0.95 ± 0.01 | 96.5 ± 0.47 | 97.1 ± 0.63 | 0.9909 ± 0.008 | 99.31 ± 0.6 | 99.33 ± 0.62 |
| DARecNet-BS | 0.95 ± 0.01 | 96.19 ± 0.98 | 97.15 ± 0.54 | 0.9918 ± 0.009 | 99.376 ± 0.699 | 99.31 ± 0.7 |
| BS-Net-Conv | 0.95 ± 0.01 | 96.39 ± 0.6 | **97.45 ± 0.48** | 0.9943 ± 0.01 | 99.5674 ± 0.25 | 99.72 ± 0.117 |
| DRL | 0.92 ± 0.01 | 94.06 ± 0.91 | 93.86 ± 0.83 | 0.9911 ± 0.0046 | 99.3261 ± 0.35 | 99.35 ± 0.36 |
| PCA | 0.93 ± 0.01 | 94.74 ± 0.39 | 95.3 ± 0.67 | 0.9732 ± 0.012 | 97.968 ± 0.91 | 98.01 ± 1.01 |
| SpaBS | 0.92 ± 0.01 | 93.68 ± 0.46 | 94.29 ± 0.54 | 0.9940 ± 0.002 | 99.545 ± 0.226 | 99.58 ± 0.2 |
| SNMF | 0.91 ± 0.02 | 93.42 ± 1.39 | 93.03 ± 1.68 | 0.9919 ± 0.0123 | 99.3929 ± 0.92 | 99.435 ± 0.81 |

Table 3: Overall Accuracy (OA), Average Accuracy (AA), and Kappa Results for the PaviaU dataset with 3 and 8 bands selected.

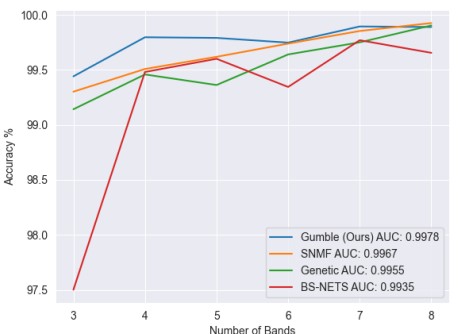 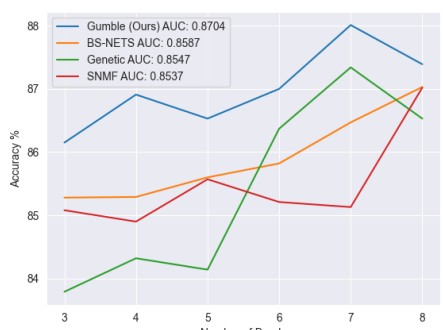

(a) Overall accuracy and AUC results for Chikusei.  (b) Mean IOU for HSI-Drive-V2 with 5 classes

Figure 4: Results for (a) Chikusei and (b) HSI-Drive-V2 datasets

## 5.2 Autonomous driving results

Mean IOU results for the drive dataset with a 5-class classification are shown in Figure 4b. As can be seen, our method clearly outperforms all other methods for all target number of bands as hence also in overall AUC. Table 4 shows detailed results for the drive dataset for 3 bands and 8 band target selection comparing our proposed method to others over 12 different evaluation metrics. As can be seen in the table, our method outperforms all other methods in all evaluation metrics when 3 bands are selected and in all but 2 metrics when 8 bands are selected (for these two cases out method is ranked closely as second best with a low margin).

| Method/Metric | 3 bands | | | | 8 bands | | | |
|---|---|---|---|---|---|---|---|---|
| | SNMF | BSNETS | Genetic | Ours | SNMF | BSNETS | Genetic | Ours |
| Mean IoU | 85.08 | 85.28 | 83.79 | **86.15** | 87.02 | 87.03 | 86.53 | **87.39** |
| Overall IoU | 92.99 | 93.24 | 92.41 | **93.59** | 94.06 | 94.08 | 93.83 | **94.24** |
| Weighted IoU | 93.53 | 93.76 | 93.01 | **94.04** | 94.48 | 94.50 | 94.33 | **94.66** |
| Mean Precision | 90.18 | 90.34 | 88.72 | **91.15** | 90.91 | 91.40 | 91.01 | **91.59** |
| Mean Recall | 92.63 | 92.88 | 93.00 | **93.35** | **94.55** | 93.91 | 93.56 | 94.18 |
| Overall Accuracy | 96.37 | 96.50 | 96.06 | **96.69** | 96.94 | 96.95 | 96.82 | **97.03** |
| Mean Accuracy | 92.63 | 92.88 | 93.00 | **93.35** | **94.55** | 93.91 | 93.56 | 94.18 |

Table 4: Detailed results for drive dataset with 3 and bands 8 selected

## 6 Analysis and Discussion

The results clearly demonstrate that our method outperforms existing state-of-the-art (SOTA) methods, particularly when selecting a small number of bands. We attribute the superior performance

of our method to its ability to learn a specific set of bands tailored to the given task, which may sacrifice information irrelevant to the task but results in better-focused feature selection. Our band selection layer requires a certain number of learnable parameters, which is the total number of bands multiplied by the number of target bands. As a future direction, simplifying the band selection layer to reduce the number of learnable parameters could make the optimization process more efficient, especially for large-scale applications.

## 6.1 SELECTED BANDS ANALYSIS

Figure 5a compares the bands selected by our method to those selected by other methods. Reconstruction-based methods tend to select bands that include representative from the extremes of the spectrum, as they optimize for data reconstruction. However, our method, which focuses on optimizing for a specific task, shows that these extreme bands are not always necessary, leading to improved task-specific performance without the need for these difficult-to-acquire bands.

Figure 5b shows the selected bands for the HSI-Drive dataset using our method for different target numbers of bands. The selection is consistent, with the top 3 bands (top of figure) remaining largely unchanged as the number of target bands increases to 8 (bottom row of figure).

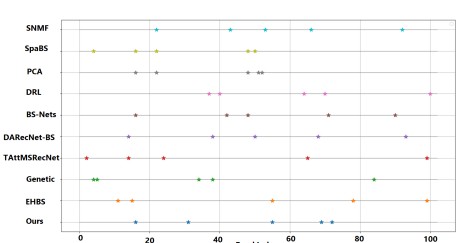

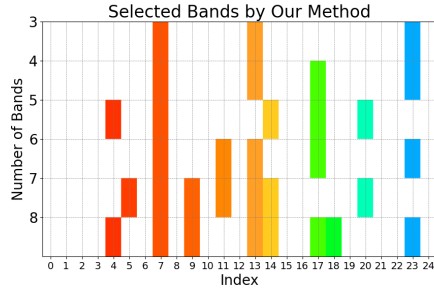

(a) Selected 5 bands of PaviaU by different models     (b) Selected bands for drive 5 class with our method

Figure 5: Illustration of the bands selected by (a) different models for PaviaU and (b) our method for HSI-Drive-V2 with different numbers of target bands.

## 6.2 INITIALIZATION

During initial experiments, we observed that the Gumbel encoder often selected duplicate bands, which reduced the diversity of the selected bands. Figure 6 illustrates the progression of band selection on the Salinas data set with a target of 6 bands. Figure 6a captures the progression of band selection given a Xavier uniform random initialization of the logits matrix. As can be seen, as the learning progresses, the selection of bands converges to only 3 different ones. To address this issue, we introduced a non-uniform initialization of the logits matrix, segmenting the band spectrum based on the target number of bands (see section 4.5). Figure 6b illustrates that this improved initialization results in the selection of 8 diverse bands.

Our novel initialization scheme of the logits matrix provided significant improvements in overall accuracy while avoiding band duplication. This was achieved by initializing each selector to focus on a different region of the spectrum, enhancing the diversity of selected bands and ultimately improving model performance. Notably, even though the initialization segmented the spectrum, the learning process was not restricted to selecting a single band in each segment. For example, Figure 7 shows the selected bands and the equivalence initialization areas for the drive dataset when selecting 8 bands from 25. As can be seen, our method did not select any band from the first segment (bands 0-2) and did select two bands in a different region (bands 12-14). We would like to note that we also tried other initialization techniques, including one where we initialized with a higher value the band indices that were selected in other methods, such as SNMF SpaBs and BSNets. This seeding performed similarly though not as well as our segment-based initialization. As our method performed better and does not rely on running other methods we chose it for our setting of choice.

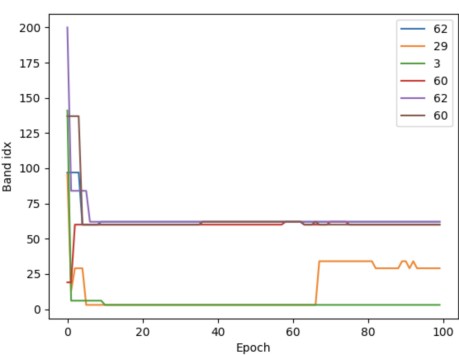
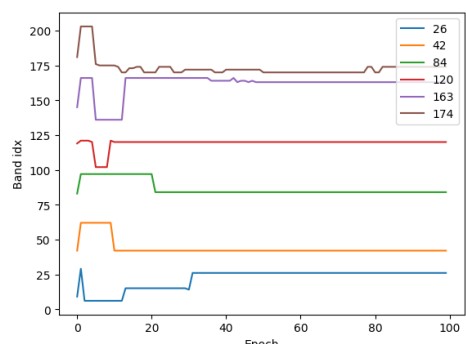

(a) Concrete encoder with random initialization, progression of band selection with 6 bands Salinas on Salinas

(b) Concrete encoder with our initialization, progression of band selection with 6 bands Salinas on Salinas

Figure 6: Progression of band selection during learning with (a) random initialization and (b) our proposed initialization



Figure 7: Selected bands and the equivalence initialization areas for drive dataset with 8 bands

### 6.3 LIMITATIONS

Our experiments required extensive hyperparameter tuning, and we observed that the model's performance was highly sensitive to these settings. Additionally, different tasks required distinct hyperparameter configurations, which highlights a limitation in terms of generalizability and ease of deployment across varied scenarios.

### 6.4 CONCLUSION

We introduced a novel supervised band-selection-as-a-layer method for deep learning models, which can be seamlessly integrated into various architectures. Our approach consistently demonstrated strong performance across all tested datasets, particularly excelling when selecting a small number of spectral bands. This capability is crucial for practical applications where reducing the number of bands allows for the use of simpler, more affordable cameras in additional real-world scenarios, such as small robot navigation, medical devices, drones and agricultural applications.

### AUTHOR CONTRIBUTIONS

To be completed in the final version

### ACKNOWLEDGMENTS

To be completed in the final version

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

# A APPENDIX

| Method/Metric | Kappa | OA | AA |
|---|---|---|---|
| Gumbel (Ours) | 0.9987 ± 0.001 | 99.8891 ± 0.10 | 99.442 ± 0.91 |
| BS-Net-Conv | 0.9960 ± 0.0051 | 99.6545 ± 0.448 | 98.99 ± 0.01 |
| SNMF | 0.9919 ± 0.001 | 99.9252 ± 0.103 | 99.56 ± 0.63 |
| Genetic | 0.9988 ± 0.001 | 99.9033 ± 0.1112 | 99.2874 ± 0.95 |

Table 5: Chikusei with 8 bands selected

| Method/Metric | Kappa | OA | AA |
|---|---|---|---|
| Gumbel (Ours) | 0.9935 ± 0.0016 | 99.44 ± 0.145 | 96.45 ± 1.68 |
| BS-Net-Conv | 0.9710 ± 0.006 | 97.5010 ± 0.5251 | 96.5618 ± 0.93 |
| SNMF | 0.9919 ± 0.003 | 99.3014 ± 0.292 | 95.644 ± 1.08 |
| Genetic | 0.9901 ± 0.001 | 99.1416 ± 0.1564 | 95.363 ± 0.846 |

Table 6: Chikusei with 3 bands selected

| Method/Metric | Kappa | OA | AA |
|---|---|---|---|
| Gumbel (Ours) | $0.9923 \pm 0.006$ | $99.31 \pm 0.53$ | $99.67 \pm 0.304$ |
| EHBS | $0.9908 \pm 0.0045$ | $98.184 \pm 0.6228$ | $97.975 \pm 0.694$ |
| genetic | $0.9921 \pm 0.0039$ | $98.12 \pm 0.86$ | $97.904 \pm 0.96$ |
| TAttMSRecNet, | $0.9838 \pm 0.0076$ | $98.549 \pm 0.68$ | $99.42 \pm 0.28$ |
| DARecNet-BS | $0.9886 \pm 0.0057$ | $99.376 \pm 0.699$ | $99.31 \pm 0.7$ |
| BS-Net-Conv | $0.9937 \pm 0.003$ | $98.4747 \pm 0.7409$ | $99.37 \pm 0.003$ |
| DRL | $0.9844 \pm 0.0074$ | $98.6078 \pm 0.666$ | $99.35 \pm 0.29$ |
| PCA | $0.9657 \pm 0.0027$ | $96.9233 \pm 0.246$ | $98.633 \pm 0.26$ |
| SpaBS | $0.9752 \pm 0.83$ | $97.783 \pm 0.748$ | $99.03 \pm 0.46$ |
| SNMF | $0.9889 \pm 0.003$ | $99.008 \pm 0.33$ | $99.56 \pm 0.1$ |

Table 7: Salinas with 8 bands selected

