# OpenReview forum: "Supervised Band Selection with a Concrete Layer for Hyperspectral Imagery in Remote Sensing and Autonomous Driving"
_ICLR.cc/2025/Conference — ICLR 2025 Conference Withdrawn Submission_

### Official Review · Reviewer_TrPK · 2024-10-31

**Soundness:** 2
**Presentation:** 3
**Contribution:** 2
**Rating:** 5
**Confidence:** 4

**Summary:**

The manuscript presents a state-of-the-art supervised band selection technique employing a concrete layer and the Gumbel-Softmax re-parameterization trick. This method stands out for its ability to integrate seamlessly into deep learning models, allowing for the dynamic selection of the most informative spectral bands specific to the task at hand. The approach excels in reducing data dimensionality while preserving critical information, which is pivotal for efficiency in remote sensing applications.

**Strengths:**

Advancing Hyperspectral Imaging Applications:
The significance of this work lies in its potential to advance hyperspectral imaging applications by enabling more efficient and effective band selection. This is particularly important for real-world applications where data volume and processing time are critical constraints.

Impact on Autonomous Driving and Remote Sensing:
The paper's contribution is significant for the fields of autonomous driving and remote sensing. By improving band selection, it can lead to more accurate and robust object identification and tracking in these domains.

Potential for Broader Adoption:
The proposed method's ability to integrate seamlessly with existing deep learning architectures suggests a high potential for broader adoption across various hyperspectral imagery applications, including environmental monitoring and medical imaging.

**Weaknesses:**

1. While the paper discusses the computational complexity of the operations involved in the proposed method, there might be concerns about scalability, especially for very large hyperspectral datasets or in real-time applications where processing speed is critical.
2. The paper does not explicitly address how the method performs under noisy conditions or with hyperspectral data that exhibits high variability. Understanding the robustness of the band selection process in such scenarios is important for real-world applications.
3. The method's focus on selecting a small number of bands might lead to the exclusion of important bands that, while not immediately informative, could provide valuable context when combined with other bands. This potential oversight should be considered.

**Questions:**

1.Why is the Gumbel-Softmax reparameterization technique suitable for addressing the problem you have proposed? This should be analyzed in detail within the introduction section.
2.The author points out in the last paragraph of the introduction that the proposed method focuses on semantic segmentation tasks, which contradicts the downstream tasks mentioned above, as downstream tasks refer to more tasks that include not only semantic segmentation tasks.
3.To my knowledge, the band selection technique proposed in the paper "Novel Gumbel-Softmax Trick Enabled Concrete Autoencoder with Entropy Constraints for Unsupervised Hyperspectral Band Selection" also employs the Gumbel-Softmax technique. Please explain in your paper the similarities and differences between your approach and the one presented in that work. Additionally, discuss the advantages of your method compared to theirs.
4.The module you proposed should be applied to more downstream tasks that have already achieved excellent performance to verify that your proposed method can select the optimal frequency band.

---

### Official Review · Reviewer_5sXA · 2024-11-02

**Soundness:** 2
**Presentation:** 2
**Contribution:** 2
**Rating:** 5
**Confidence:** 3

**Summary:**

The paper proposes an embedded supervised band selection method for hyperspectral imagery (HSI) using a concrete selector layer based on the Gumbel-Softmax re-parameterization trick. This method, integrated directly into the model’s training pipeline, aims to simplify the band selection process without pre-processing, achieving adaptive, task-specific feature selection. The authors tested the approach on four datasets, spanning remote sensing and autonomous driving, claiming improved efficiency and model performance compared to existing methods.

**Strengths:**

1. The introduction of a concrete selector layer, specifically adapted for HSI band selection, provides an innovative solution, especially in the context of autonomous driving.
2. Integrating band selection directly into the training pipeline is a key strength, potentially improving task-specific adaptability and reducing the need for separate preprocessing steps.
3. The paper presents experiments across four datasets, including both remote sensing and autonomous driving data, which supports its claims of versatility and task-specific performance.

**Weaknesses:**

1. **Lack of Clarity in Methodological Justification**: The choice of the Gumbel-Softmax re-parameterization trick and the concrete selector layer is not fully justified in terms of why these methods are better suited for HSI band selection compared to other techniques. Further discussion on why this choice is optimal, given HSI’s specific demands, would strengthen the theoretical basis.

2. **Over-Reliance on Baseline Comparisons Without Detailed Analysis**: While the authors compare their method against nine baselines, there is limited exploration of why their method outperforms others beyond reporting metrics. For example, analyzing why the Gumbel-Softmax layer improves performance in some datasets but not in others could provide a clearer understanding of the approach’s strengths and limitations.

3. **Inadequate Explanation of Initialization Techniques**: The paper introduces an initialization scheme that reportedly prevents duplicate band selection. However, the description lacks depth, particularly regarding its comparative advantage over other initialization techniques. Furthermore, more quantitative results could better substantiate the claim that this initialization scheme improves performance.

4. **Limited Insight into Computational Efficiency**: Although embedded methods are generally more efficient, the authors do not provide detailed computational or runtime analyses to substantiate efficiency claims. Reporting metrics such as training time, memory usage, or computational complexity relative to baseline methods would be valuable, especially for real-time applications like autonomous driving.

5. **Insufficient Discussion on Hyperparameter Sensitivity**: The method reportedly requires extensive hyperparameter tuning, which raises concerns about its practical utility. Despite noting the sensitivity, the authors do not provide a systematic study of how specific hyperparameters (e.g., temperature \(\tau\) in Gumbel-Softmax) affect performance across datasets, leaving questions about the method’s robustness.

6. **Generalization Limitations**: The paper acknowledges that different tasks require different hyperparameter configurations but fails to address this as a limitation. Without a standardized approach or guidelines for parameter selection, the applicability across tasks and scalability of this method remains unclear.

7. **Ambiguity in Experimental Settings**: Although the datasets are well described, the details on experimental protocols are sparse. For example, it is unclear if the training-validation splits were consistent across all experiments, which could affect reproducibility and comparability of results.

**Questions:**

see weeknesses

---

### Official Review · Reviewer_6Sp4 · 2024-11-03

**Soundness:** 2
**Presentation:** 1
**Contribution:** 1
**Rating:** 3
**Confidence:** 5

**Summary:**

The paper presents an embedded method for supervised band selection in hyperspectral images that addresses the challenges of data dimensionality and the adaptability of current band selection methods. This approach uses a “concrete selector layer” based on the Gumbel-Softmax reparameterization, which enables task-specific selection of the optimal bands, eliminating the need for preprocessing. The method can be integrated into downstream models and has been evaluated on four hyperspectral data sets with applications in remote sensing

**Strengths:**

- The paper introduces a unique plug-and-play, embedded approach for supervised band selection using a concrete selection layer with the Gumbel-Softmax trick, directly integrated within the model's training pipeline, which is relatively rare in hyperspectral band selection research.

- By dynamically selecting bands based on task requirements, this approach could reduce data preprocessing and streamline the pipeline for real-world application integration.

- The method is tested on both remote sensing and autonomous driving datasets, offering insights into its generalizability across different hyperspectral imagery applications

**Weaknesses:**

- There are few details about the proposed method. It is difficult for readers to understand the specific processes and details of the algorithm, considerably weakening the contribution of this work.

- The paper highlights the need for extensive hyperparameter tuning for each dataset and task, including parameters such as the temperature and noise factors (see Section 4.5). For instance, the optimal temperature value τ is selected from a range of [0, 10], with different optimal settings for remote sensing (τ=1.5) and autonomous driving (τ=8.5) datasets. This intensive tuning process is time-consuming and limits the model’s scalability, making it less practical for new tasks without considerable retuning.

- In Section 4.5, the authors report that the Gumbel-Softmax concrete layer often selects duplicate bands when randomly initialized, resulting in reduced diversity among selected bands (Figure 6a). To mitigate this, the authors introduced a custom initialization method that segments the spectrum and assigns different bands to each selector, avoiding duplication (Figure 6b). However, this approach complicates the implementation and may not be easily replicable in other scenarios, particularly if the spectral characteristics differ.

- The model shows substantial variability in performance across tasks, as discussed in Section 6, indicating that task-specific tuning is necessary. For example, remote sensing tasks use a 3D CNN architecture, while the autonomous driving task relies on a UNet-based framework. This dependence on different architectures reduces the model’s transferability to other applications, particularly in cases where the band selection task or network structure differs significantly.

- As described in Sections 4.2 and 4.3, the model’s evaluation heavily relies on 3D CNN and UNet architectures for the remote sensing and autonomous driving tasks, respectively. While effective for segmentation, this approach may not generalize well to other network structures or machine learning tasks, such as classification, thus limiting the method’s adaptability for broader applications.

- Although the model has been shown to be suitable for real-time applications, the paper does not include practical deployment scenarios, such as on-site or on-device evaluations. This omission leaves open questions about the practical feasibility and performance of the method outside of controlled experimental settings, which would be important for validating its usability in applications such as autonomous driving or environmental monitoring.

**Questions:**

- The description of methodology is too rough. Providing a figure or some kind of processing framework will help readers understand it better.
- Could you pls explain the criteria and procedure you used for tuning the hyperparameters across different datasets? Specifically, how were final values for parameters chosen, and to what extent did these settings impact generalizability across tasks? According to the write up, it seems that the selection of parameters depends entirely on manual tuning, and the inability to achieve an adaptive approach will greatly reduce the universality of the algorithm.
- Could you pls elaborate on how frequently duplicate band selection occurred during standard initialization? Additionally, would the proposed spectrum-segmentation initialization method generalize well to datasets with significantly different spectral distributions?
- Given the reliance on 3D CNN and UNet architectures, have you considered using other networks that are more efficient and with greater adaptability?
- Could you pls provide more details on how the selected bands impacted downstream performance, particularly in cases where fewer bands were chosen? Are there any specific spectral regions that consistently contribute to better model performance across datasets?
- Have you conducted any tests and evaluations in practical scenarios to verify the robustness of the algorithm in real-world situations, such as real-time processing under limited computational resources or low-power environments?

---

### Official Review · Reviewer_V9Fb · 2024-11-04

**Soundness:** 2
**Presentation:** 2
**Contribution:** 1
**Rating:** 3
**Confidence:** 5

**Summary:**

The paper proposes a supervised band selection method based on the concrete layer to address the issue of high data dimensionality in hyperspectral images. This method introduces Gumbel-Softmax reparameterization, enabling dynamic selection of optimal bands during training without additional preprocessing, achieving seamless integration with downstream models. Evaluations on four hyperspectral datasets, covering three remote sensing benchmark tasks and one autonomous driving task, demonstrate the method's effectiveness.

**Strengths:**

The proposed band selection method is simple and effective, enhancing the downstream models' performance when the number of bands is limited.

**Weaknesses:**

1. The authors should provide a framework diagram or algorithm flowchart to enhance the readability of the paper and assist readers in understanding.
2. There is a significant overlap between the content of Section 3 "Methods" in this paper and EHBS [1], including the task setup section. It is recommended that the authors write based on their own research rather than directly copying existing work.
3. Compared to EHBS [1], the primary contribution of this paper is the application of the Gumbel-Softmax [2] reparameterization technique for band selection, which focuses on application rather than theoretical and technical innovation.
4. The paper mentions that the experiments required extensive hyperparameter tuning, and different tasks needed distinct configurations, indicating a lack of robustness and generalization ability of the method. Current deep learning models do not heavily rely on band selection when learning downstream tasks, and the high cost of hyperparameter tuning raises concerns about the practicality of the method.
5. The authors should supplement experimental results from other models on downstream tasks, including traditional algorithms and deep learning models (based on CNN and Transformer architectures), to demonstrate the applicability of the method.
6. The authors should add comparisons with newer band selection algorithms from the past two years, such as MOCS-BS [3] and IROA-BS [4].
7. Given that the method introduces additional parameters, it is suggested to supplement the experimental results regarding the number of parameters, training time, GFLOPs, etc.
References:
[1] Zimmer Y, Glickman O. Embedded hyperspectral band selection with adaptive optimization for image semantic segmentation[J]. arXiv preprint arXiv:2401.11420, 2024.
[2] Jang E, Gu S, Poole B. Categorical reparameterization with gumbel-softmax[J]. arXiv preprint arXiv:1611.01144, 2016.
[3] Ou X, Wu M, Tu B, et al. Multi-objective unsupervised band selection method for hyperspectral images classification[J]. IEEE Transactions on Image Processing, 2023, 32: 1952-1965.
[4] Jia H, Li Z. Hybrid Multistrategy Remora Optimization Algorithm-Based Band Selection for Hyperspectral Image Classification[J]. IEEE Transactions on Geoscience and Remote Sensing, 2024.

**Questions:**

Does the proposed method not only select entire bands but also allow for the selection of specific values within the bands based on task requirements?

---

### Author Response · Authors · 2024-11-22
**Note on withdrawal**

We would like to thank the reviewers for the valuable feedback on our submission. As we were not able to address the reviewers' comments in the short time of the rebuttal period, we have decided to withdraw this paper and will address them in a revised version. We plan to submit the updated paper to an appropriate upcoming venue.

---

### Note · Authors · 2024-11-22

**Comment:**

We would like to thank the reviewers for the valuable feedback on our submission. As we were not able to address the reviewers' comments in the short time of the rebuttal period, we have decided to withdraw this paper and will address them in a revised version. We plan to submit the updated paper to an appropriate upcoming venue.

**Withdrawal Confirmation:**

I have read and agree with the venue's withdrawal policy on behalf of myself and my co-authors.